# The Relative Importance of Human Disturbance, Environmental and Spatial Factors on the Community Composition of Wetland Birds

**Seid Tiku Mereta** [1,*,†], **Pieter Lemmens** [2,†], **Luc De Meester** [2], **Peter L. M. Goethals** [3] **and Pieter Boets** [3,4]

1   Department of Environmental Health Science and Technology, Jimma University, Jimma P.O. Box 378, Ethiopia
2   Laboratory of Aquatic Ecology, Evolution and Conservation, KU Leuven, Ch. Deberiotstraat 32, 3000 Leuven, Belgium; pieter.lemmens@kuleuven.be (P.L.); luc.demeester@kuleuven.be (L.D.M.)
3   Laboratory of Environmental Toxicology and Aquatic Ecology, Department of Animal Sciences and Aquatic Ecology, Ghent University, Coupure Links 653, Building F, 9000 Ghent, Belgium; peter.goethals@ugent.be (P.L.M.G.); pieter.boets@oost-vlaanderen.be (P.B.)
4   Provincial Centre of Environmental Research, Godshuizenlaan 95, 9000 Ghent, Belgium
*   Correspondence: seidtiku@yahoo.com; Tel.: +251-913296056
†   These authors contributed equally to this work.

**Abstract:** The present study investigates the relative importance of human disturbance, local environmental and spatial factors on variations in bird community composition in natural Ethiopian wetlands with high biodiversity conservation value. We quantified bird abundances, local environmental variables and human disturbances at 63 sites distributed over ten wetlands in two subsequent years. Variation partitioning analyses were used to explore the unique and shared contributions of human disturbance, local environmental variables and spatial factors on variations in community compositions of wetland bird species. Local environmental variables explained the largest amount of compositional variation of wetland bird species. Productivity-related variables were the most important local environmental variables determining bird community composition. Human disturbance was also an important determinant for wetland bird community composition and affected the investigated communities mainly indirectly through its effect on local environmental conditions. Spatial factors only played a minor role in variations in bird community composition. Our study highlights the urgent need for integrated management approaches that consider both nature conservation targets and socio-economic development of the region for the sustainable use and effective conservation of wetland resources.

**Keywords:** community composition; human disturbance; local environment; space; wetland birds

## 1. Introduction

Wetlands are defined as areas that are transitional between terrestrial and aquatic systems, where the water table is usually at or near the surface, or the land is covered by shallow water [1]. The three important characteristics that are associated with and used to constitute a wetland include hydrology, hydrophytic vegetation and hydric soils [2]. Wetlands perform a wide variety of ecological functions, including provisioning of habitat for wildlife, purification of catchment surface water, floodwater attenuation, groundwater recharge, climate regulation and erosion control [1,3]. Furthermore, wetlands play a vital role in providing a wide range of ecosystem services for millions of people, mainly those living in low-income countries [3–5]. Wetlands are increasingly recognized for their high contribution to biodiversity [6,7]. Wetlands provide crucial habitats for many species since they comprise multiple microhabitats that provide a variety of resources. The high habitat heterogeneity of wetlands typically results in highly diverse bird communities,

often including rare and endemic species [8]. The species composition of bird communities is increasingly used to assess the ecological status of habitats, including wetlands [9,10].

Habitat degradation and modification have been recognized as major anthropogenic disturbances underpinning the current loss of biodiversity worldwide [11,12]. Wetlands are particularly vulnerable to human pressure as they are often located in densely populated areas [13]. The rapid growth of the human population in many countries results in an increase in human demands, which has led to strong reclamation of intact wetland areas. Consequently, over half of the wetlands worldwide have currently been destroyed or are severely affected by agriculture, mining practices and urban development [14]. As a result, many wetland bird species have experienced a profound decline in population density over the last decades. Among these, rare and endangered species are especially vulnerable, since their populations are relatively small and often restricted to a limited number of localities [15]. Several studies indicated that habitat loss, habitat fragmentation and land-use intensification strongly contribute to the overall decline of bird populations [15,16].

Although studies have demonstrated that bird community composition in a variety of ecosystems is determined by both deterministic environmental processes and dispersal-driven stochastic processes [17–20], the relative importance of both factors is still a topic of scientific debate [21]. For example, Gianuca et al. [20] found that local environmental factors are more important in determining compositional variation in a Brazilian coastal bird community than spatial factors, whereas Guadagnin et al. [19] indicated that bird species composition is also influenced by spatial factors. On the other hand, land use intensity was found to be an important variable determining bird species composition in Chile [22].

Although several studies have demonstrated the importance of the local environment, human disturbance and spatial factors on bird community composition, few investigations distinguish the relative importance of each explanatory factor [20,23]. One of the difficulties is that these variables are often highly correlated with each other, which hampers the correct evaluation of their relative importance. Ignoring these potentially confounding effects may result in misleading inferences about the impact of individual explanatory variables on community characteristics [17], and thus, may undermine correct biodiversity conservation management decisions [24]. Understanding the relative importance of local environmental conditions, spatial factors and human disturbances on variation in bird community composition is thus crucial for the development of effective conservation programs.

Ethiopia has a rich avifauna with more than 926 bird species, of which 21 are endemic and 19 are globally threatened [6]. Seventy-three hotspots have been identified as important bird areas in Ethiopia, of which 30 sites comprise wetlands [6,25]. Despite the high importance of these wetlands for biodiversity conservation, their management is poorly addressed. Rapid human population growth triggers the expansion of agricultural and urban areas and also promotes additional exploitation activities in wetland areas [26]. As a result, several wetlands either disappeared or are currently on the verge of disappearing [13,26], while others are prone to severe habitat degradation. This may have profound effects on the community composition of wetland birds, but also on the ecosystem services that they are providing to humans. Identifying the factors shaping bird communities in human-altered wetlands is pivotal for the development of management strategies that aim for the protection of Ethiopia's rich avifauna.

The present study investigates the bird community composition in ten natural wetlands in southwest Ethiopia. Our overall aim is to explore the relative importance of local environmental variables, human disturbance and spatial factors on variations in the wetland bird community composition. Our specific objectives are: (1) to quantify the proportion of variation in bird species composition explained by local environmental variables, human disturbances and space; (2) to compare the relative importance of these explanatory variables; and (3) to identify the major environmental variables and human disturbances determining bird species composition. Our study fills in important knowledge gaps and identifies the mechanisms by which human disturbances affect wetland bird communities.

Such information is important to support science-based management programs and policy decisions that aim to stop the ongoing loss of biodiversity. Our findings can contribute to the conservation of the ecological integrity of East African wetlands.

## 2. Materials and Methods

### 2.1. Study Area

This study was conducted in ten natural wetland systems in the Gilgel Gibe I watershed situated in Southwest Ethiopia (Figure 1). The Gilgel Gibe I watershed has an area of approximately 5125 km$^2$ at its confluence with the great Gibe river and comprises relief hills and mountains with an average elevation of 1700 m above mean sea level. The regional climate is wet, with an average annual rainfall of approximately 1550 mm. Precipitation follows a bi-modal pattern with a wet season from June till early September and a dry season between December and January. The mean annual air temperature is 19 °C. During recent decades, the Gilgel Gibe I watershed has been subjected to considerable human pressure, which mainly originates from high human population growth, agricultural expansion, water resources development and ongoing urbanization. Currently, the watershed largely consists of agricultural land.

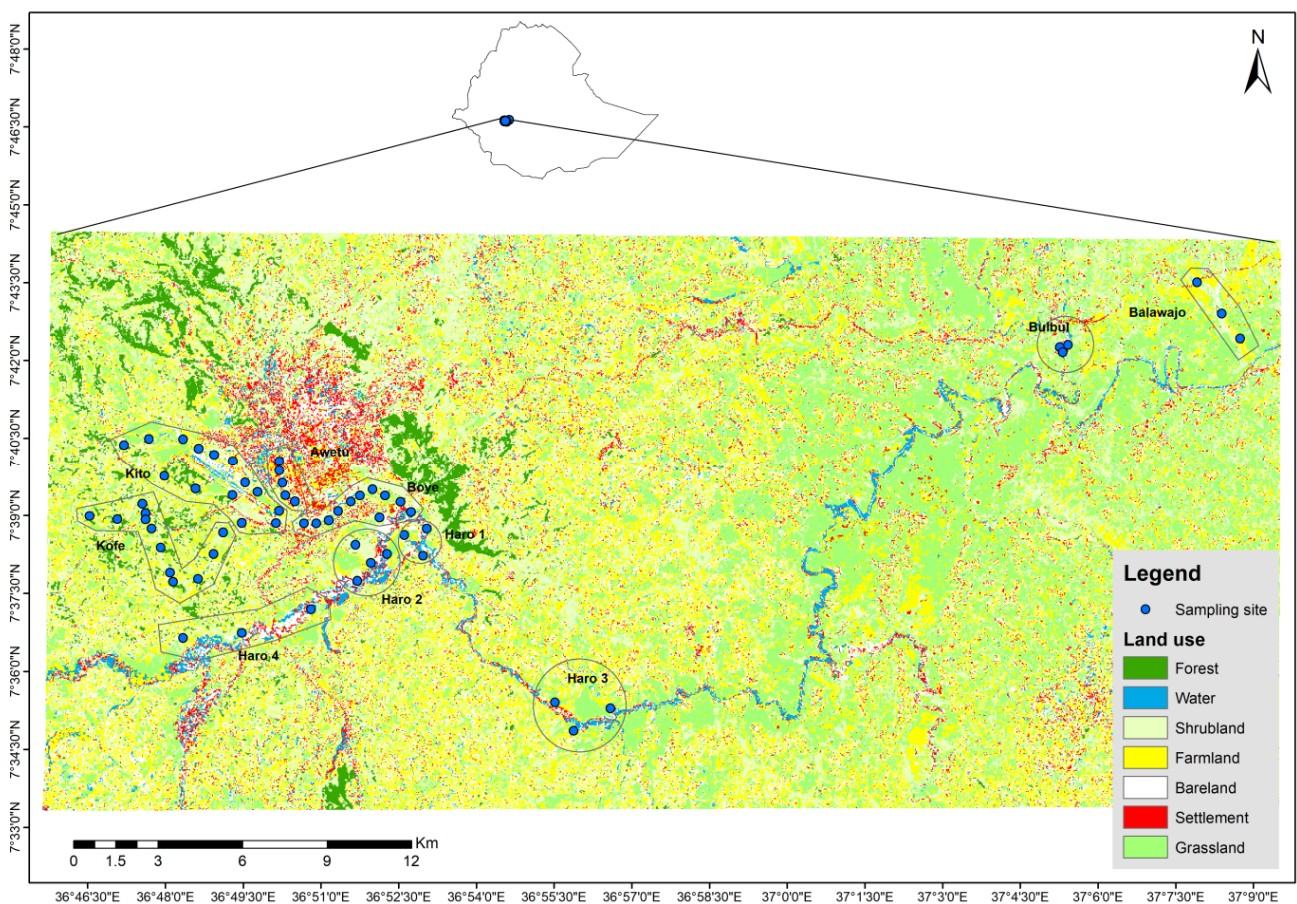

**Figure 1.** Location of the study area and wetland sampling stations in the Gilgel Gibe watershed, Southwest Ethiopia.

The set of investigated wetlands consists of four permanent riverine wetlands (Kofe, Kitto, Boye and Awetu), four temporary floodplain wetlands (Haro 1 to 4), one semi-permanent floodplain wetland (Bulbul) and one temporary riverine wetland (Balawajo). Permanent wetlands contain water throughout the entire year, while semi-permanent wetlands hold water for a relatively short time period (approximately two to three months after the end of the rainy season). Temporary wetlands only hold water during and immediately after the end of the rainy season (less than two months). Riverine wetlands

are hydrologically connected to a river or stream, while floodplain wetlands are located adjacent to a river or stream that overflows periodically. We selected 63 observation sites distributed over the ten wetlands. Sites were selected within each wetland along a gradient of visible human disturbance (from nearly non-impacted to heavily disturbed sites, e.g., presence of waste dumping, clay mining, crop farming, etc.) [27]. The number of observation sites within each wetland depended on wetland size (ranging from 5 to 110 hectares), with smaller wetlands having a lower number of sampling sites than larger wetlands (see Table S1).

### 2.2. Bird Surveys

Bird surveys took place in each wetland at each observation site during the dry and wet seasons of 2010 and 2011. Each site was visited four times over a period of two years. Bird abundance data were collected using a point-count technique within a 100 m radius [28]. Point counts are commonly used to estimate bird community composition and diversity [29]. All birds were visually located and identified to species level using binoculars ($10 \times 42$, Kite) during 15 min between 07:00 and 10:00 am or between 4:00 and 6:00 pm local time because that is when bird activity is the highest. The moment of observation (morning or late afternoon) was randomized across sites within each wetland. Scientific nomenclature and bird taxonomy followed Sibley and Monroe [30]. The geographic coordinates of each bird observation site were recorded using a hand-held global positioning system unit (GPS) (Garmin GPS 60, Garmin international Inc., and Olathe, KS, USA).

Observed bird species were classified into three distinct groups based on their association with wetlands following Almaw [31]: (1) wetland-dependent specialists, (2) wetland-dependent generalists and (3) wetland-associated species. Wetland-dependent specialist species are those that fully depend on aquatic habitats for nesting, feeding and roosting. Examples include ducks, cormorants and grebes. Wetland generalists are species that are frequently found in wetlands but are sometimes also seen in other habitats as well, such as ibises, some weavers, warblers, and plovers. Wetland-associated birds are mainly found in nearby upland habitats but are attracted to wetlands for foraging. Examples are Abyssinian ground hornbill, African wattled lapwing, African mourning dove and African paradise monarch (see also Table S2).

### 2.3. Environmental Variables

Environmental variables were quantified at each observation site in each wetland. Water depths and thicknesses of the sludge layer were measured at each site using a graduated stick. Conductivity, pH, daytime dissolved oxygen concentration and water temperature were measured in situ in the field using a multi-probe meter (HQ30d Single-Input Multi-Parameter Digital Meter, Hach Company, Loveland, USA). In vivo chlorophyll a concentration was used as a proxy for phytoplankton biomass and was measured in the field using a hand-held fluorometer (AquaFluor, Turner Designs, Sunnyvale, California, USA). A water sample (200 mL) was taken from each site and subsequently filtered through 0.45 μm filter paper in the field for the determination of nitrate, ammonia and orthophosphate concentrations. Unfiltered water (500 mL) was used to determine the five-day biochemical oxygen demand ($BOD_5$), chemical oxygen demand (COD), total organic nitrogen (TON) and total phosphorous (TP) concentrations in the laboratory. Water samples were kept in a cold box and transported to the laboratory for further analysis. Nitrate and ammonia were analyzed according to the American Public Health Association Standard methods [32]. Total phosphorus samples were digested in a block digester using ammonium persulfate and sulfuric acid reagent [32]. Both soluble reactive phosphorous and total phosphorous were analyzed using the stannous chloride method [32]. Biochemical oxygen demand ($BOD_5$) was measured according to the standard method as described in APHA [31]. Samples for COD and TON were also digested and measured with photometric kits (HACH LANGE) using a Hach DR5000 spectrophotometer. The percentage of vegetation cover

(emergent, floating and submerged) was visually estimated within a 100 m radius around each observation site [33].

### 2.4. Human Disturbance

The extent of different types of human disturbance was determined at each observation site in each wetland during the dry and wet seasons of both sampling years. Human disturbances were categorized into multiple categories: crop cultivation, waste dumping, clay mining, grazing, tree plantation, vegetation clearance, ditching, filling and draining. The proximity and magnitude of each of these disturbances at each observation site were quantified on an ordinal scale following Mereta et al. [27] (1 = no or minimal; 2 = moderate; and 3 = high). The overall human disturbance for each site was calculated by summing the individual disturbance values of human activities (nine different activities in total) (see Table S3 for more details).

### 2.5. Spatial Variables

Spatial variables were generated based on the geographical coordinates of the observation sites using principal coordinates of neighbor matrices (PCNM), as described by Borcard and Legendre [34]. PCNM analysis allows for the detection of spatial structures across a wide range of geographical scales [34]. The geographical coordinates of the observation sites were used to construct a Euclidean distance matrix, which was subsequently truncated at the smallest distance that keeps all sites connected in a single network. The truncated Euclidean distance matrix was used in a principal coordinate analysis (PCoA) to extract eigenvectors associated with positive eigenvalues to be used as explanatory variables in further statistical analyses.

### 2.6. Data Analyses

We used separate variation partitioning analyses to determine the unique and shared contributions of local environment conditions, human disturbance and spatial variables to the variation in the composition of the entire bird community and each category of birds (wetland specialist, wetland generalist and wetland associated species). Variation partitioning analysis allows partitioning of the total amount of variation explained by a statistical model into unique and shared contributions of sets of explanatory variables [35,36].

We first conducted redundancy analysis (RDA) to evaluate the overall effect of environment, human disturbances and space on the composition of the entire bird community and each category of birds. Significant variables within each set of explanatory variables were subsequently identified using forward selection based on the adjusted $R^2$ double-stopping criterion [37]. The association of bird community composition with significant explanatory variables was visualized using ordination plots of principal component analyses (PCA) [36].

Secondly, we applied a variation partitioning analysis based on partial RDA to assess the relative importance of unique and shared contributions of significant sets of explanatory variables (environment, human disturbance and space) to variations in the composition of the entire bird community and each category of birds [35,36]. We also conducted an additional variation partitioning analysis, including wetland identity as an explanatory variable to explore the potential importance of unmeasured wetland-specific conditions. The interpretation of a significant unique contribution of a variable set is straightforward and indicates a direct effect, independent of the other explanatory variable sets in the model. The shared contribution among explanatory variables could occur as a result of the indirect effects of one variable over the other explanatory variable. For example, when the explained variation is shared between local environment and human disturbance, this may indicate an effect of human disturbance through its impact on the measured environmental variables. Similarly, shared explained variation between the local environment and space likely reflects an effect of spatially-structured environmental conditions. For the interpretation of the importance of wetland identity, we focused on the fraction of variation uniquely

explained by wetland identity, which basically comprises the importance of unmeasured wetland-specific variables.

We used mean values across seasons and years for environmental variables and bird community composition for the statistical analyses. All environmental variables, except pH, were logarithmically transformed to improve the normality of the data. Bird abundance data were Hellinger transformed [38]. The significance of RDA models was assessed with Monte-Carlo permutations (*n* = 999). All statistical analyses were conducted in R (version 3.1.1, R Development Core Team, 2013) using the pcnm, rda and varpart functions of the vegan library [35,39].

## 3. Results

### 3.1. Bird Surveys

A total of 9654 individuals belonging to 140 species, 54 families and 15 orders of birds were recorded. The most dominant order was the Passeriformes, consisting of 21 families and 58 species, including the Long-billed pipit (*Anthus similis*), White-tailed swallow (*Hirundo megaensis*) and Abyssinian longclaw (*Macronyx flavicollis*), which are considered as near-threatened species. The second dominant order was Charadriiformes, consisting of five families and 11 species, including the Black-tailed godwit (*Limosa limosa*), listed as a near-threatened species. The observed order Gruiformes included three families and seven species, including the Wattled crane (*Bugeranus carunculatus*) and Black-crowned crane (*Balearica pavonina*), which are listed as vulnerable species and Rouget's rail (*Rougetius rougetii*), which is an endemic species and listed as near threatened. Among the 140 species, 83 species could be categorized as wetland associated, 17 as wetland-dependent generalists and the remaining 40 as wetland-dependent specialist birds. Some bird species such as Hadada Ibis (*Bostrychia hagedash*), Sacred ibis (*Threskiornis aethiopicus*) and African wattled lapwing (*Vanellus senegallus*) were recorded in all wetlands. However, the majority of the bird species were found in a limited number of wetlands. For example, the Blue-breasted kingfisher (*Halcyon malimbica*), Giant kingfisher (*Megaceryle maxima*) and lesser moorhen (*Gallinula angulata*) were recorded only in the Boye wetland. Fulvous whistling duck (*Dendrocygna bicolor*) was found only in Bulbul and Kito wetlands. On the other hand, the Glossy Ibis (*Plegadis falcinellus*) was recorded in the Bulbul and Kito wetlands only. See Table S2 for a complete species list.

### 3.2. The Effect of Environment, Human Disturbance, Space on Bird Community Composition

RDA results revealed that local environmental variables, human disturbances, space and wetland identity explained a significant proportion of the variation in community composition of each category of birds (specialists, generalists and wetland associated) and to the entire bird community (Tables 1 and 2). Forward selection identified different sets of significant local environmental variables for each category of birds. Overall, the productivity-related variables, such as pH and nutrient concentration, seemed to have a positive effect on the abundance of the majority of bird species (Figure 2). In addition, habitat permanency negatively affected the community composition of wetland-dependent generalist birds. With regard to human disturbance, clay mining had an important negative effect on specialist and generalist bird species. Several generalist species were also negatively affected by vegetation clearing, grazing and wetland drainage (Figure 2). Wetland-associated bird species were positively associated with grazing, while farming, vegetation clearing and the occurrence of plantations seem to have a negative effect on the majority of observed species. Forward selection identified different sets of spatial descriptors for specialist, generalist and wetland-associated communities, but the selected spatial descriptors overall corresponded with broader spatial scales.

**Table 1.** Results of redundancy analyses separately testing for the effect of local environment, human disturbance, space and wetland identity on the community composition of wetland specialists, wetland generalists and wetland-associated birds. See supplementary material S1 for abbreviations of local environmental variables.

| | | df | Variance | F | $R^2_{adj}$ | P | Selected Variables * |
|---|---|---|---|---|---|---|---|
| specialists ($n = 40$) | environment | 16 | 0.199 | 1.36 | 0.085 | 0.003 | pH, oxygen concentration |
| | land-use | 9 | 0.136 | 1.645 | 0.086 | 0.001 | clay mining |
| | space | 6 | 0.105 | 1.897 | 0.080 | 0.001 | PCNM3, PCNM5 |
| | wetland ID | 10 | 0.153 | 1.697 | 0.101 | 0.001 | |
| generalist ($n = 17$) | environment | 16 | 0.249 | 3.557 | 0.398 | 0.001 | pH, sludge, $NO_3$, TON, chla, temperature, $PO_4$, TP, $NH_4$, EC, permanency |
| | land-use | 9 | 0.153 | 3.056 | 0.223 | 0.001 | plantation, clearing, grazing, clay mining, drainage |
| | space | 6 | 0.141 | 4.256 | 0.234 | 0.001 | PCNM 1, PCNM2, PCNM3, PCNM5 |
| | wetland ID | 10 | 0.256 | 6.994 | 0.492 | 0.001 | |
| associated ($n = 83$) | environment | 16 | 0.194 | 1.905 | 0.190 | 0.001 | TP, vegetation, $NO_3$, temperature, $PO_4$, BOD, chla, water depth |
| | land-use | 9 | 0.104 | 1.608 | 0.081 | 0.001 | plantation, farming, grazing, clearing |
| | space | 6 | 0.089 | 2.076 | 0.094 | 0.001 | PCNM1, PCNM3, PCNM4 |
| | wetland ID | 10 | 0.172 | 2.843 | 0.230 | 0.001 | |

\* Explanatory variables selected based on forward selection following [37].

**Table 2.** Results of redundancy analyses separately testing for the effect of local environment, human disturbance, space and wetland identity on the entire wetland bird community composition.

| Variables | df | Variance | F | $R^2_{adj}$ | *p* | Selected Variables * |
|---|---|---|---|---|---|---|
| environment | 19 | 0.264 | 2.240 | 0.275 | 0.001 | water depth, sludge, temperature, TON, $NO_3$, TP, morphology, permanency, chla, vegetation cover |
| land use | 9 | 0.146 | 2.236 | 0.152 | 0.001 | drainage, grazing, clearing, plantation, farming, clay mining, waste dumping |
| space | 6 | 0.124 | 2.837 | 0.151 | 0.001 | PCNM1, PCNM3, PCNM4, PCNM5 |
| wetland ID | 10 | 0.220 | 3.669 | 0.301 | 0.001 | |

\* Explanatory variables selected based on forward selection following [37].

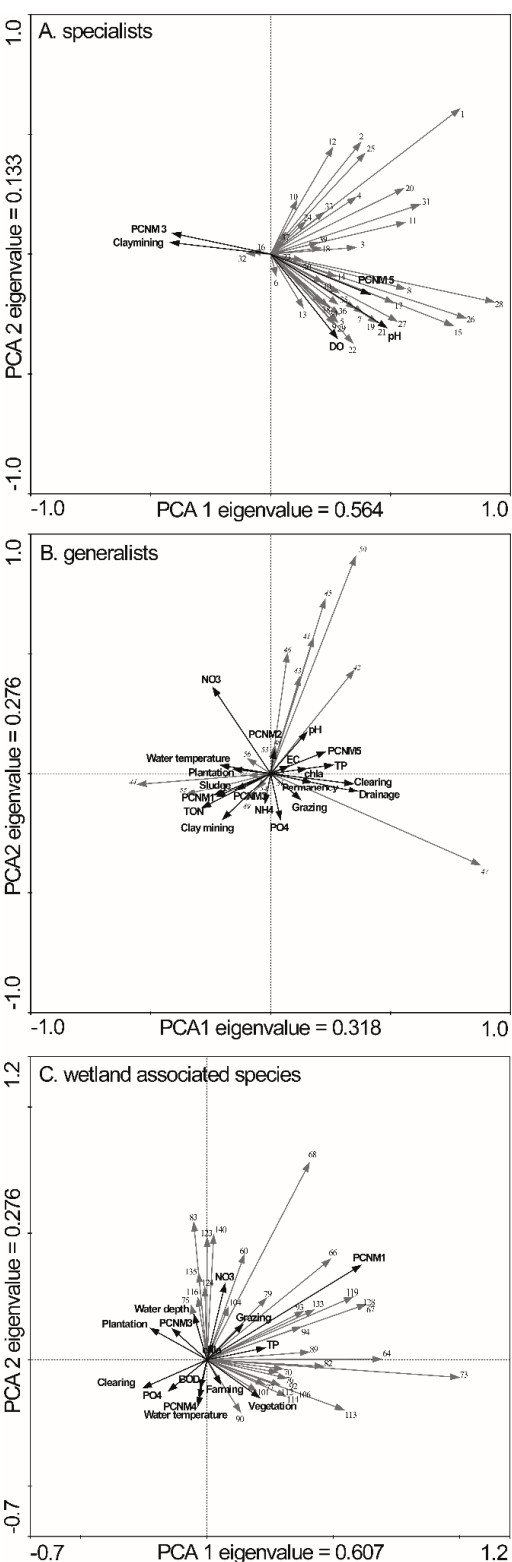

**Figure 2.** Ordination plot of a principal component analysis on the community composition of (**A**) wetland specialist, (**B**) wetland generalist and (**C**) wetland-associated bird species. Grey arrows indicate the different species (see Table S2 for the species corresponding to the numbers). Black arrows represent significant explanatory variables from each set of explanatory variables (environment, human disturbance and space) and were plotted as supplementary information to not affect the ordination. For reasons of clarity, only species that occurred in at least 5% of the observation sites were visualized for wetland-associated birds.

### 3.3. The Relative Importance of Local Environment, Human Disturbance, Space and Wetland Identity on Bird Community Composition

Variation partitioning analyses revealed that the order of importance of local environment, human disturbance and space in explaining variation in community composition among observation sites was similar for each category of birds (Figure 3). Local environmental variables explained the largest amount of compositional variation of wetland specialist, wetland generalist and wetland-associated bird species (7.5%, 39% and 15.8%, respectively), followed by space (6.9%, 23.6% and 8.3%, respectively) and human disturbance (5%, 21.7% and 7.4%, respectively). Likewise, the variation partitioning analyses performed for the entire bird community indicated that the environment tends to be relatively more important in explaining compositional variation in bird communities. Human disturbance and spatial factors also affect bird community composition mainly indirectly through the environment (Figure 4). Human disturbance had a relatively small but significant unique effect, especially on wetland generalist bird community composition. A relatively large proportion of compositional variation explained by local environmental variables in each category of birds was shared with human disturbance and space. In addition, local environmental variables also had a considerable unique contribution to the explained compositional variation. This fraction was relatively higher for wetland-associated birds (approximately half of the total variation explained by local environment) than for wetland generalists and wetland specialists (approximately one-fourth of the total variation explained by local environment). The amount of variation explained by space was mainly shared with local environmental variables, and with both local environmental variables and human disturbance. However, space also had a significant unique effect on the compositional variation of wetland generalist and wetland-associated bird species. The unique effect of space became insignificant when wetland identity was included in the variation partitioning analysis (Figure S1). Wetland identity determined compositional variation in bird communities largely due to the shared contributions with local environmental variables, human disturbance and space. However, wetland identity uniquely explained a significant fraction of compositional variation in generalist and wetland-associated bird communities.

**A.** Wetland specialist community

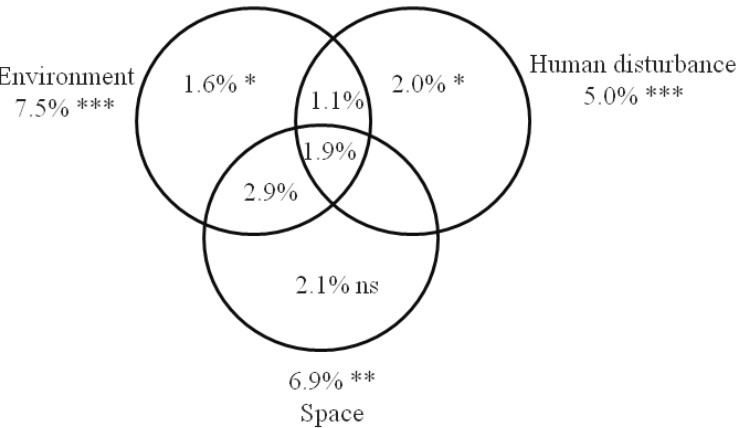

**Figure 3.** *Cont.*

**B.** Wetland generalist community

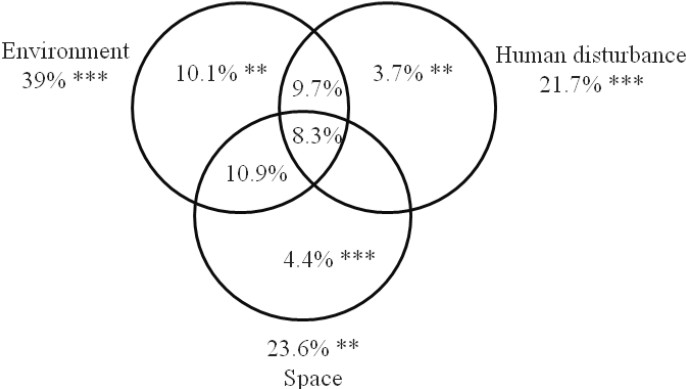

**C.** Wetland-associated community

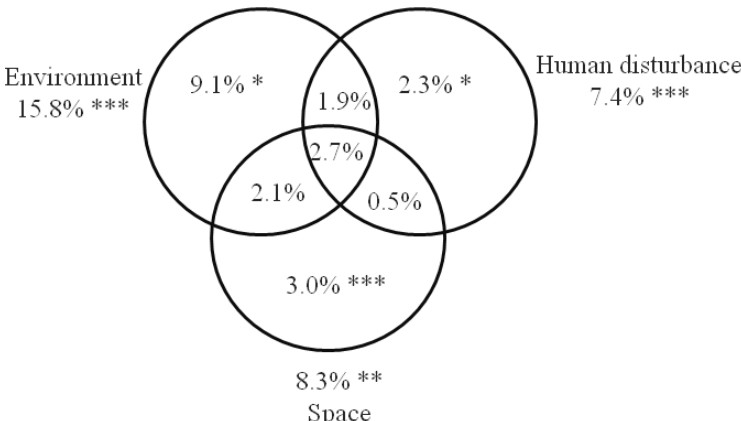

**Figure 3.** Results of variation partitioning analyses on the community composition of (**A**) wetland specialist, (**B**) wetland generalist and (**C**) wetland-associated birds with local environmental variables (environment), human disturbance and space as explanatory variables. Percentages within the diagrams represent the adjusted $R^2$. $R^2_{adj}$ values < 0 are not mentioned. Asterisks denote the significance level (* $p < 0.05$, ** $p < 0.01$, *** $p < 0.001$). The significance of shared fractions could not be tested.

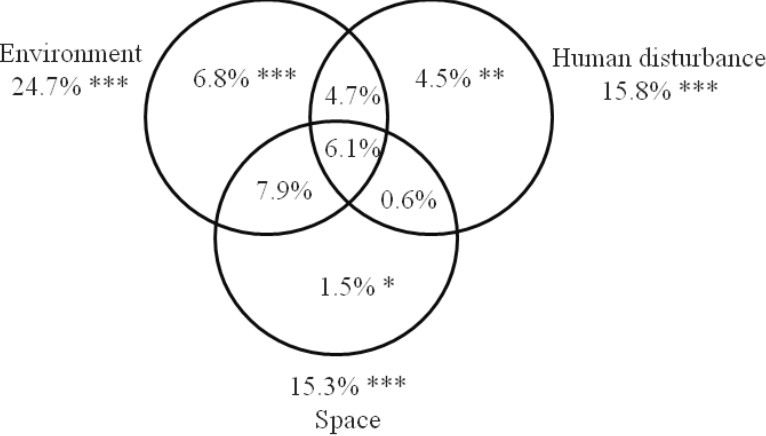

**Figure 4.** Venn-diagram showing the results of a variation partitioning analysis based on the entire wetland bird community with environment, human disturbance and space as explanatory variables. Percentages within the diagrams represent the adjusted $R^2$. $R^2_{adj}$ values < 0 are not mentioned. Asterisks denote the significance level (* $p < 0.05$, ** $p < 0.01$, *** $p < 0.001$). The significance of shared fractions cannot be tested.

## 4. Discussion

The present study investigates the relative importance of local environmental variables, human disturbance and spatial factors on the community composition of wetland birds in a set of ten natural wetlands in southwest Ethiopia. Overall, our analyses indicate that variation in community composition of wetland birds is largely determined by local environmental variables and human disturbance, while spatial factors played a minor role.

Local environmental variables explained the largest proportion of the variation in community composition in each category of birds (wetland generalist, wetland specialist and wetland associated). This finding is consistent with earlier studies highlighting the importance of local habitat conditions for bird community characteristics. For example, Gianuca et al. [20] report that local environmental variables are important in explaining compositional variation in Brazilian coastal bird communities. Similarly, Tozer et al. [15] reported that local environmental variables played a role in wetland bird abundance. Although the unique contribution of local environmental variables was significant for each category of wetland birds, the largest fraction of the variation explained by local environmental variables was shared with space, human disturbance and a combination of space and human disturbance. Shared variation between environment and space indicates that at least some important environmental variables were spatially structured, whereas the shared explained variation between local environment and human disturbance likely refers to an indirect effect of human disturbance by altering local environmental conditions. Shared explained variation between environment, space and human disturbance suggests that some spatially structured environmental variables are also determined by human disturbances. When wetland identity was taken into account in the variation partitioning analyses, the unique effect of environment became insignificant, and the variation explained by the environment was shared with wetland identity. This implies that at least some environmental factors such as water depth and vegetation cover were wetland specific. For example, the floodplain wetlands Haro and Bulbul were characterized by high productivity, whereas riverine wetlands were characterized by high vegetation cover. The fraction uniquely explained by the wetland identity fraction suggests the importance of unmeasured wetland-specific conditions.

In the present study, different sets of significant environmental variables were identified for each category of birds. However, productivity-related variables, such as chlorophyll a, pH and nutrient concentrations, overall seemed to be the most important variables determining variation in community composition in wetland specialist and generalist species among sites (Figure 2). Rajpar et al. [8] indicated that wetland productivity is an important predictor for the variation in community composition of wetland birds in Malaysia. In our study area, floodplain wetlands are highly productive [13], mainly due to the long residence time of water, which facilitates the sedimentation of suspended solids and increases the total solar irradiance available for phytoplankton growth in the water column [40]. In addition, these habitats are also used for crop cultivation and grazing of cattle during the dry season, which contributes to the accumulation of nutrients [13] and in turn affects wetland productivity and prey abundance for birds. The availability of prey items, such as fish and invertebrates, is known to influence bird abundance and species composition [10,41].

In this study, spatial factors alone played a minor role in explaining bird species composition. The lack of a strong, pure spatial effect might be related to the relatively small geographical scale of our study area. Considering the high mobility of most bird species and their sensitivity to subtle changes in environmental conditions, it is not surprising that spatial factors played a minor role in the variation in bird community composition. However, our findings are in contrast with Guadagnin et al. [19], who found evidence for the effects of space on bird species composition in fragmented wetlands of Southern Brazil. One explanation for this discrepancy might be the difference in geographical scale between the two studies. Our study reveals that human disturbance is an important factor underpinning variation in wetland bird community composition. Effects of human disturbance were largely mediated through its effect on local environmental variables. Among

the human disturbances, vegetation clearing and clay mining were important disturbances affecting the compositional variation of specialist and generalist bird communities. The clearing and removal of wetland vegetation negatively affect bird communities by reducing food availability and destroying habitats for roosting and nesting [42,43]. Indeed, several earlier studies clearly demonstrate that the removal of vegetation affects neo-tropical bird communities through habitat loss, reduced forest-patch size, as well as by lowering connectivity between different populations and by increasing interactions with species from surrounding non-forest patches [44,45]. However, not all bird species are equally vulnerable to vegetation clearance [46]. Several studies indicated that habitat specialists are more susceptible to habitat alterations compared with generalists [47,48]. Generalist species use various habitat types in the landscape matrix and are less affected by habitat fragmentation than specialists, which are more dependent on one or few habitat types [49].

Clay mining for brick manufacturing, one of the major human disturbances, is expanding rapidly in the investigated wetlands due to the increasing urbanization in the study region. In addition to removing the topsoil, brick making is also considered an important cause of vegetation clearing, as it uses large amounts of wood from wetland riparian habitats to burn bricks [33,50]. Interestingly, we observed a positive effect of cattle grazing on some wetland generalist and wetland-associated birds. This finding is consistent with the study of Soka et al. [51], who showed that the abundance of some bird species increases in the presence of low to moderate cattle grazing. For example, Kour and Sahi [52] show that the abundance of cattle egrets increased with increasing livestock density due to the increased availability of insects and other prey items. In addition, grazers can increase bird foraging efficiency by providing vantage points [53].

## 5. Conclusions

Our analyses provide quantitative information on the relative importance of human disturbance, environmental and spatial factors on variation in community composition of wetland birds in Ethiopia. We found that local environmental variables explained most of the variation in community composition of different groups of wetland birds, while spatial factors only played a minor role. The amount of variation explained by environmental variables was relatively low for wetland specialist birds. This might be attributed to unmeasured wetland-specific environmental variables, biotic interactions or stochastic processes. We, therefore, suggest that future studies attempt to include wetland-specific environmental variables (e.g., water depth), biotic factors (e.g., predation and competition) and stochastic factors to better understand the governing factors that affect the community composition of wetland bird species in the study area. On the other hand, human disturbances, such as clay mining and vegetation clearing, had a negative effect on the compositional variation of specialist and generalist bird communities, mainly through their effect on local environmental variables such as habitat conditions and water quality. Overall, the degradation of wetlands by human activities negatively affected bird species composition. Therefore, integrated management approaches that consider both nature conservation targets and socio-economic development of the region are urgently needed for the sustainable exploitation of the Ethiopian wetlands.

**Supplementary Materials:** The following are available online at https://www.mdpi.com/article/10.3390/w13233448/s1, Figure S1: Results of variation partitioning analyses on the community composition of (A) wetland specialist, (B) wetland generalist and (C) wetland associated birds with environment (E), human disturbance (H), space (S) and wetland identity (W) as explanatory variables. Table S1. Overview of key wetland characteristics and mean (minimum, maximum) values for local environmental variables for each wetland across sampling sites, seasons (wet and dry) and years (2010 and 2011). Permanency: 1 = Temporary; 2 = Semi-permanent; 3 = Permanent. Abbreviation environmental variables: TON = total organic nitrogen; $NH_4^+$ = ammonium; $NO_3^-$ = nitrate; TP = total phosphorus; $PO_4^{3-}$ = orthophosphate; $BOD_5$ = Five day Biochemical Oxygen Demand; COD = Chemical Oxygen Demand; EC = Electric Conductivity; pH = logarithmic measure of hydrogen ion concentration. Table S2: List of the observed bird species with their relative abundance

and frequency of occurrence across all observation sites. Table S3: Criteria and ordinal scale used to quantify the magnitude of different types of human disturbance in close proximity of each sampling site.

**Author Contributions:** S.T.M. and P.L. conceived the main idea of the paper, collected the data and wrote the first draft of the manuscript. P.L. and P.B. analyzed the data and contributed to the writing of the manuscript. P.L.M.G. and L.D.M. contributed actively to the discussions and edited the manuscript. All authors have read and agreed to the published version of the manuscript.

**Funding:** This research was funded by VLIR-UOS. STM is a recipient of an IUC (Institutional University Cooperation) PhD scholarship from VLIR to carry out this work.

**Institutional Review Board Statement:** The study protocol has been reviewed and approved by Jimma University Institutional review board (IRB) with a reference number of HRPGS/4015/2010.

**Informed Consent Statement:** Not applicable.

**Data Availability Statement:** Data used in this study will be made available on request to the corresponding author.

**Acknowledgments:** The authors would like to acknowledge VLIR-UOS for financing this study. The authors wish to thank all people who helped with field data collection.

**Conflicts of Interest:** The authors declare that they have no competing interests. The funder of the study had no role in the study design, data collection, data analyses, data interpretation or writing of the manuscript.

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
