# Peer review of "The Relative Importance of Human Disturbance, Environmental and Spatial Factors on the Community Composition of Wetland Birds"

_water, doi:10.3390/w13233448_

Round 1

Reviewer 1 Report

Title: The relative importance of human disturbance, environmental and spatial factors on the community composition of wetland birds

The resubmitted manuscript has been improved. In some cases, the authors could not provide improvements as this would take additional research. I believe that at least the first author will monitor the conditions in the field in the future.

The authors have provided the requested analyses of the whole dataset of bird community, which is included in responses and is worth to be included. The most important thing of this Venn-diagram is the high share of “HUMAN DISTRURBANCE”, which exceeds the values calculated in case of specific meta-communities. If we want to emphasize this influence, which is generally more negative than positive this will represent a good argument.    

Author Response

Point 1: The resubmitted manuscript has been improved.

Response 1: We thank the reviewer for the positive remark and proving constructive comments which greatly improve our manuscript.

Point 2: In some cases, the authors could not provide improvements as this would take additional research. I believe that at least the first author will monitor the conditions in the field in the future.

Response 2: We thank the reviewer for this suggestion. We have recognized the need for follow-up study to understand the extent and trends of wetland loss and bird community composition in the study area. Currently, the first author is looking for financial support from Ethiopian basin authority, ecohydrology project to collect data on wetland size, water quality, land-use, macroinvertebrate and bird community composition. This will allow for conducting wetlands trend analyses and suggesting management recommendations for sustainable use of wetlands in the basin.

Point 3: The authors have provided the requested analyses of the whole dataset of bird community, which is included in responses and is worth to be included. The most important thing of this Venn-diagram is the high share of “HUMAN DISTRURBANCE”, which exceeds the values calculated in case of specific meta-communities. If we want to emphasize this influence, which is generally more negative than positive this will represent a good argument.  

Response 3: We have now included the outputs of the analyses done on the entire bird community (Table 2 and Figure 4). We have also provided information on the analyses of data on the entire bird community in the methods section and description of the findings under results section of the revised manuscript.

Reviewer 2 Report

The current version is improved, specific figures

But needs further polishing by native English speaker

Author Response

Point 1 The current version is improved, specific figures

Response 1: We thank the reviewer for the positive remark and proving constructive comments which greatly improve our manuscript.

Point 2: But needs further polishing by native English speaker

Response 2: We have checked the entire manuscript and corrected the spelling and grammatical errors.

Reviewer 3 Report

Authors of the ms took into consideration my suggestions and improve their text. The ms illustrates the influence on the birds communities various factors, in particular, human activity. I think it can be published.

Author Response

Point 1: Authors of the ms took into consideration my suggestions and improve their text. The ms illustrates the influence on the birds communities various factors, in particular, human activity. I think it can be published.

Response 1: We thank the reviewer for the positive remark and providing

constructive comments which greatly improve our manuscript.

This manuscript is a resubmission of an earlier submission. The following is a list of the peer review reports and author responses from that submission.

Round 1

Reviewer 1 Report

I think that the current study about wetland birds is interesting for a broad community and was done carefully. The text is logical and clear. I have a question: It was shown that all factors explain relatively low amount of variation in community composition of wetland specialist (7,43%; 6,82%; 4.91%). Can the authors explain this somehow?

Also, it will be usefull to show clearer in the main text how species communities in the studied areas differ from each other (i.e. what is the difference between wetland specialists from different areas, between wetland associated birds...).

Please, make corrections:

p. 7, line 15 - “84 species could be categorized as wetland associated”. However 81 species were mentioned as wetland associated in the table 1.

Figure 3 – please, add the explanation of the letters E, H, S in the legend (do the same in Supplements)

Reviewer 2 Report

In this work, the authors investigated the relative importance of environment, human disturbance and  space in the community composition of wetland birds in Southwest Ethiopia.

The manuscript is interesting, but needs improvements as shown below:

1-In abstract, the lines from 18 to 22 shows repeated sentences, please try to rewrite.

2-In Introduction, It is better to add a paragraph describing a definition of  wetlands, types, causes and importance.

3-In data analysis, the authors should explain the RDA and PCA analyses and why they used them.

4-In results, Fig.2, the gray lines are not clear, also the words. Please improve it.

Reviewer 3 Report

The manuscript deals with wetlands, which are definitely worth to be studied and protected. There was a large amount of data collected in this study about ten years ago. I think that a lot of readers would like to know the situation now -if it has changed for worse or for better.  

This data set could provide clearer results. Variation partitioning analyses enable us to determine the “net” influences of separate factors or groups of factors. In this sense it is not correct and reasonable to use the proportions calculated as “gross” effects of these three groups – values without subtraction of the shares of correlating factors which blurs the picture. Such net values are significantly lower than gross values (e.g. 1.61% instead of 7.43%) and point at relative “un-importance” of these factors on the compositions of bird community. Net values representing net effects also change the order of the importance of specific groups. What is surprising is very low proportion of environmental factors/conditions (1.6%) on the composition of specialists, since the specialist species are usually most susceptible to environmental conditions and so is the composition of this meta-community.

I suggest to use the net values in your results and discussion and explain or discuss the net effects. I suggest major changes of the mscr.

Have you performed analyses with the entire wetland bird community instead of these three sub-communities? It would be interesting to see these results as well.

Minor comments:

I suggest slightly different title: The relative importance of environment, human disturbance and space on the composition of wetland bird community.

Abstract:

Ln.15, 16 and elsewhere in the text:  I suggest to write more definite expressions like environmental parameters instead of “environment” and spatial factors instead of “space”. (e.g. human disturbance, environmental and spatial factors)

Ln. 18 to 23: I suggest to join the third and fourth sentence into single one to avoid the repetition.

Methods:

Page 5, Ln 4: riparian habitats are also wetlands – please redefine this category of birds

Results:

Figure 2: these figures are not readable – the labels are to small

Figure 3: Please reduce the figures to one decimal place, since these are already percent values. Such values would also be more readable.

Variation partitioning analyses enable us to determine the “net” influences of separate factors or groups of factors. In this sense it is not correct and reasonable to use the proportions calculated as “gross” effects of these three groups – values without subtraction of the shares of correlating factors which blurs the picture. Such net values are significantly lower than gross values (e.g. 1.61% instead of 7.43%) and point at relative “unimportance” of these factors on the compositions of bird community. Net values representing net effects also change the order of the importance of specific groups. What is surprising is very low proportion of environmental factors/conditions (1.6%) on the composition of specialists, since the specialist species are usually most susceptible to environmental conditions and so is the composition of this meta-community.